# Efficacy of Lavage by Tube under Local Anesthesia versus Arthroscopic Treatment of Acute Septic Arthritis of Native Knee

**DOI:** 10.3390/diagnostics13030371

**Published:** 2023-01-19

**Authors:** Byung Hak Oh, Youn Moo Heo, In UK Yeo, Woo Jin Shin, Hyun Jin Yoo

**Affiliations:** 1Department of Orthopedic Surgery, Konyang University College of Medicine, Daejeon 35365, Republic of Korea; 2Myunggok Medical Research Institute, Konyang University, Daejeon 35365, Republic of Korea

**Keywords:** knee, septic arthritis, arthroscopic treatment, local anesthesia, tube insertion, lavage by tube

## Abstract

Although arthroscopic treatment is a minimally invasive surgery that effectively treats septic arthritis of the knee joint, it requires general or regional anesthesia. This study aimed to compare the clinical results of lavage after tube insertion versus arthroscopic treatment. Patients treated with arthroscopic treatment were included in group I (*n* = 76), while those treated with lavage by tube were included in group II (*n* = 34). We investigated the following in all patients: demographics, underlying disorders, initial serum white blood cell (WBC) count, C-reactive protein (CRP) level, synovial fluid WBC and polymorphonuclear cell counts, causative organism, initial Kellgren–Lawrence grade, lavage number, interventional delay, hospitalization days, CRP normalization time, and Western Ontario McMaster Universities Osteoarthritis index scores for clinical outcomes at 3 months postoperative. The mean interventional delay was significantly greater in group I (23.6 ± 15.6 h vs. 8.7 ± 9.3 h, *p* < 0.001). The lavage by tube featured a significantly shorter interventional delay time than arthroscopy, while the CRP decrease rate did not differ between groups. Moreover, lavage by tube showed no significant differences in outcomes, including laboratory results and functional outcomes at 3 months postoperative.

## 1. Introduction

Septic arthritis (SA) is an arthropathy caused by infection with microorganisms. SA may occur during hematogenous spreading, wounds, trauma, surgery, acupuncture, and injection. The knee joint, which is most commonly involved, has an increasing incidence [1,2]. SA can lead to sequelae such as joint structural changes and joint malfunction due to severe inflammatory reactions. Furthermore, it can result in a life-threatening condition with a mortality rate of approximately 3–29%. Thus, SA requires prompt treatment including drainage, lavage, and intravenous antibiotics [3,4].

The gold standard for diagnosing the septic origin is the isolation of pathogenic causative organisms; however, blood or synovial fluid Gram-staining and culturing have poor yields and take time. Moreover, if the culture fails, the initiation of antibiotics can negatively affect microbiological samples [5]. However, SA can be life-threatening and require prompt treatment. Thus, the diagnosis of SA was made comprehensively using clinical symptoms, laboratory results, and findings of pathogenic causative organisms [6]. This information is usually used for diagnosing SA based on laboratory findings such as white blood cell (WBC) count, erythrocyte sedimentation rate, C-reactive protein (CRP) level in the blood, WBC from the synovial fluid, and the rate of polymorphonuclear leukocytes from the synovial fluid [6]. Although the CRP response is nonspecific for diagnosing SA, it has never been used as a single diagnostic tool but is used to aid the diagnosis of SA. CRP testing is cost-effective and easily performed in most hospitals; therefore, CRP is a well-established marker of inflammation and infection. It is usually used to screen for infectious activity and antibiotic efficacy [7].

The treatment of SA requires intravenous antibiotics and prompt intervention, such as open arthrotomy, arthroscopic treatment, and drainage using a syringe. Several studies have reported the effectiveness of arthroscopic treatment. Arthroscopic treatment is minimally invasive versus open arthrotomy as an initial treatment and results in better postoperative rehabilitation [1,8,9]. Despite its advantages, infection recurrence or aggravation rates were 10–50% in previous studies [1,10,11]. However, it is difficult to repeat arthroscopic treatment in such cases since it requires general or regional anesthesia.

Old age, the patient’s general condition, a pre-existing joint disease, delayed treatment, and bacteria isolated from the knee are factors associated with a poor prognosis. Acute SA usually has one or more poor prognostic factors [12]. The confirmations of operability and fasting time are necessary for general or regional anesthesia. Therefore, arthroscopic treatment delays are inevitable for the confirmation of operability and fasting time and can contribute to the poor prognosis of SA [1,12].

The lavage method using a tube inserted under local anesthesia does not require further anesthesia after the tube insertion. Therefore, it is possible that lavage without delay is not influenced by the patient’s general condition. This study aimed to compare the clinical results of lavage after tube insertion and arthroscopic treatment. This study hypothesized the following: first, lavage by tube would involve a shorter interventional delay; and second, it would have comparable outcomes, including laboratory results.

## 2. Materials and Methods

Between January 2014 and December 2018, patients with SA were treated arthroscopically and included in group I (*n* = 76). Patients diagnosed with SA between January 2019 and April 2022 were treated with lavage tubes and included in group II (*n* = 16). The inclusion criterion was SA of the knee joint as confirmed by laboratory markers in the blood and joint fluid obtained by arthrocentesis. The diagnosis of SA was supported by the Newman criteria [6] and the patient’s clinical symptoms. The clinical symptoms were as follows: systemic fever, tenderness of the entire joint, swelling, effusion, joint irritability, heating sensation, erythema, and limited motion. An elevated CRP level, serum leukocytosis, and an elevated synovial fluid WBC count and rate of polymorphonuclear leukocytes were used to diagnose SA. According to Newman [6], we included patients who had positive causative organisms, positive culture-associated hematogenous, and negative synovial fluid and blood cultures; however, pus or turbid fluid was present or the histological and laboratory evidence of SA was noted. Our detailed laboratory diagnostic criteria for SA were as follows: elevated serum CRP with WBC > 50,000/µL, polymorphonuclear leukocyte rate <75%, and synovial fluid glucose/serum glucose < 50%.

Serum laboratory results were evaluated weekly. The last follow-up day was at 3 months postoperative to confirm the resolved laboratory results and clinical symptoms. This study was approved by our local institutional review board. The study participants provided informed consent for the use of their data.

### 2.1. Evaluation Methods

To test our hypothesis, we investigated the following variables in all patients: demographics, underlying disorders (hypertension and diabetes), initial serum WBC count and CRP levels, synovial fluid WBC and polymorphonuclear cells, causative organisms, initial Kellgren–Lawrence (K-L) grade, lavage number, time to intervention, hospitalization day (length of hospital stay), and CRP normalization time. Clinical outcomes of the range of motion (ROM) at 3 months postoperative were evaluated using flexion contracture (FC) and further flexion (FF). Clinical scores were evaluated using the Western Ontario and McMaster University pain, stiffness, and function scales preoperatively and 3 months postoperatively.

The K-L grade was assessed using anteroposterior knee radiography before the intervention. The lavage number was the total number of SA interventions required for treatment. The time to intervention was defined as the time that elapsed after the diagnosis of SA. The CRP normalization time was defined as the time required to obtain the first normalized result. The FC and FF were measured at the femoral epicondyle as the knee joint while lying on the bed using a goniometer at a unit of 10°. The fully extended knee was assumed parallel to the horizontal line. The FF was defined as limited when active flexion was >100° considering the daily activity.

### 2.2. Surgical Technique

All patients with SA underwent primary arthroscopic intervention performed by an arthroscopic surgeon. Anteromedial and anterolateral portals were used as the routine arthroscopic portals, and >3 L of normal saline was used for the lavage. We performed a synovectomy and debridement of the inflammatory tissue. Subsequently, a Hemovac was maintained at 4 days postoperative.

The principle of lavage using the tube is shown in Figure 1. Superomedial and superolateral portals were used for tube insertion under local anesthesia. First, a stab incision on the superolateral portal was made, mosquito forceps penetrating the joint capsule were inserted, and the portal was dilated. Second, a switching stick was inserted and a stab incision made on the superomedial portal. Third, the drain was retracted from the superomedial to superolateral portals using Kelly forceps or a beath pin. Finally, both sides of the drain were tagged to prevent falls (Figure 2). Once the tube was located, >3 L of normal saline was used for the lavage. The tube was maintained for approximately 4 days postoperative for natural drainage, and the dressing was changed daily. The tube was planned to be removed once the CRP level had decreased at approximately 4 days postoperative, but this did not occur, and an additional intervention was planned.

The same antibiotic treatment regimen was used for the two treatment courses. A first-generation cephalosporin was used as the primary antibiotic until the organism was cultured. The antibiotic was changed appropriately once the causative organism was identified on the culture. Intravenous antibiotics were used for approximately 2–3 weeks and then changed to oral antibiotics.

### 2.3. Statistical Analysis

All statistical analyses were performed using SPSS (version 22.0; IBM Corp., Armonk, NY, USA). Intra- and inter-rater reliabilities of the measurement were assessed using an intra-class correlation coefficient (ICC) analysis. Data are reported as mean and standard deviation for continuous variables. The differences in quantitative variables (age, body mass index [BMI], serum and synovial fluid analysis, initial K-L grade, lavage number, time to intervention, hospitalization, CRP normalization time, and clinical scores) were analyzed using the Mann–Whitney test. Fisher’s exact test was used to compare qualitative variables (sex, right or left side, underlying disorders, causative organisms, ratio of initial K-L grade, and ROM). Statistical significance was set at *p* < 0.05.

## 3. Results

The ICC values were satisfactory for the radiographic measurements (inter-rater reliability, >0.81; intra-rater reliability, >0.80). The demographic characteristics of the patients by study group are presented in Table 1. There were more re-aggravation cases in group I than in group II, but the difference was not significant. No statistically significant differences were found between the groups in terms of age, sex ratio, site (right or left), BMI, underlying disorder, or follow-up period.

The initial laboratory results were not significantly different between the groups (Table 2). The mean CRP level was higher in group II (16.7 ± 9.1 mg/dL) than in group I (15.0 ± 8.2 mg/dL); however, the mean serum WBC count was higher in group I (10825 ± 3560/µL) than in group II (9787 ± 2339/µL). However, the differences were not statistically significant. The mean synovial fluid WBC count was higher in group II (71804 ± 41949/µL) than in group I (70934 ± 43322/µL) by 870, but the difference was not statistically significant. In both groups, the most common causative organism was *Staphylococcus aureus* (group I, 64% (49); group II: 60% (20)).

Comparisons of patient characteristics are shown in Table 3. The initial K-L grades showed no statistically significant differences. There was no statistically significant difference in the number of lavages. The mean lavage number of group I (1.1 ± 0.3) was similar to that of group II (1.1 ± 0.4; *p* < 0.952). The time to the intervention was significantly delayed by approximately 15.9 h in group I (23.6 ± 15.6) compared to group II (6.7 ± 9.3, *p* < 0.001). There were no statistically significant differences in hospitalization or CRP normalization time. The CRP level and numerical rating scale (NRS) score change patterns are shown in Appendix A, showing no significant differences.

A comparison of the clinical outcomes revealed no significant inter-group differences (Table 4). FC occurred more frequently than FF, but there were no significant inter-group differences of prevalence. There were no cases of FC > 10° and limited FF cases in group II, versus two cases of FC > 10° and five cases of limited FF in group I. However, this difference was not statistically significant. The preoperative clinical scores were similar between groups without significant differences. There were no significant differences in the clinical scores at 3 months postoperative. However, improved clinical scores at 3 months postoperative versus preoperative were observed in both groups.

## 4. Discussion

The principal findings of this study were as follows. First, lavage by tube featured significantly less of an interventional delay as well as more opportunities for reintervention upon reaggravation. Second, lavage by tube showed no significant differences in outcomes, including laboratory results and functional outcomes at 3 months postoperative. Thus, our hypotheses were verified.

The prevalence of SA is increasing owing to societal aging, the use of intra-articular injections, and the inappropriate use of antibiotics [3,13,14,15,16]. The definitive diagnostic criteria for SA are controversial. However, SA can lead to sequelae and life-threatening conditions in a short period of time because of its aggressive progression. Therefore, if findings suggestive of SA are observed, rapid treatment significantly influences prognosis. Reducing the time to intervention and summarizing the procedure are important, especially for patients with general weakness [3,4].

Appropriate early intervention is important in patients with SA. Higher rates of mortality, osteomyelitis, osteonecrosis, severe articular destruction, and stiffness may occur if the intervention is delayed [17]. Nil per os (NPO) is one problem associated with arthroscopic treatment for SA in older patients or those with comorbidities. Furthermore, NPO for general or spinal anesthesia causes delayed intervention. Patients with SA usually have comorbidities such as immunodeficiency, diabetes mellitus, and pre-existing joint diseases. Interventions performed under local anesthesia do not require fasting, which is critical for older or high-risk patients [18,19]. In this study, when lavage under local anesthesia was used, no local or systemic complications occurred. Thus, interventions performed under local anesthesia can be considered advantageous for patients with various underlying diseases or known side effects of general anesthesia [1].

The treatment of SA with surgical drainage of the infection is highly controversial [20,21]. Mathews et al. [9] reported a lack of robust clinical evidence of SA. However, the removal of purulent material is essential for successful treatment, and washout is mainly performed during arthroscopy or arthrotomy with or without synovectomy and debridement. Tsumura et al. [22] treated knees with SA using arthrotomy. They also performed a synovectomy, debridement, and continuous irrigation. Nine of eleven patients were cured, and the average Japanese Orthopedic Association score was 84 ± 11. However, deteriorated osteoarthritis occurred in all but one patient. Stutz et al. [23] studied the arthroscopic treatment of SA using Gächter criteria [24]. Twenty-one of twenty-two stage I patients were treated with arthroscopic irrigation combined with antibiotics. However, 52% of stage II and 75% of stage III patients required additional arthroscopic irrigation procedures. Johns et al. [1] reported that arthroscopic treatment was more successful in irrigation procedures and boasted a better range of motion compared with arthrotomy. However, 50% (*n* = 60) of the 119 patients required additional arthroscopic procedures. Furthermore, 71% (*n* = 30) of the 42 patients who underwent arthrotomy required additional procedures. Most surgeons prefer arthroscopy for managing septic knees because they can be drained and the debridement defined by visualization [1]. However, consensus is lacking regarding the treatment of debridement using synovectomy [20]. In this study, drainage of SA using a tube was effective as evidenced by the decline in the CRP level and NRS score.

This study has some limitations. First, patients who underwent lavage by tube demonstrated a slower decrease in NRS scores than those who underwent general anesthesia. The patients were treated with local anesthesia when the tube was first inserted. Subsequently, the penetrated tube remained in the suprapatellar synovial cavity. This could have caused the pain in the early phase of the lavage. Second, a relatively small number of patients were treated using a lavage tube, and the statistical analyses were performed on a small number of samples. Furthermore, the delay to intervention was significantly different between groups; therefore, the results could be confirmed as statistically significant even with a relatively small number of patients. Finally, this study had a short-term follow-up period (approximately 3 months). However, considering the period required for antibiotic treatment, it was sufficient. Therefore, it was neither too soon nor too late to determine the treatment outcome.

## 5. Conclusions

The lavage by tube featured a significantly shorter interventional delay time than arthroscopy, while the CRP decrease rate did not differ between the groups. Moreover, lavage by tube showed no significant differences in outcomes, including laboratory results and functional outcomes at 3 months postoperative.

## Figures and Tables

**Figure 1 diagnostics-13-00371-f001:**
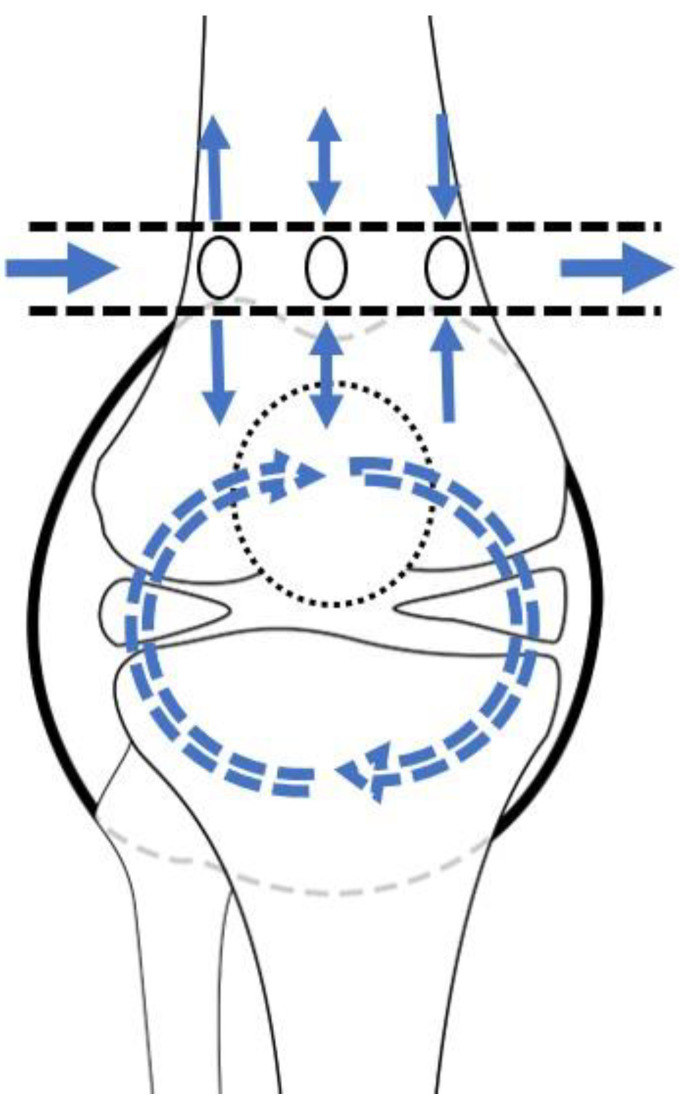
Schematic diagram of lavage by tube. The arrows depict the saline flow.

**Figure 2 diagnostics-13-00371-f002:**
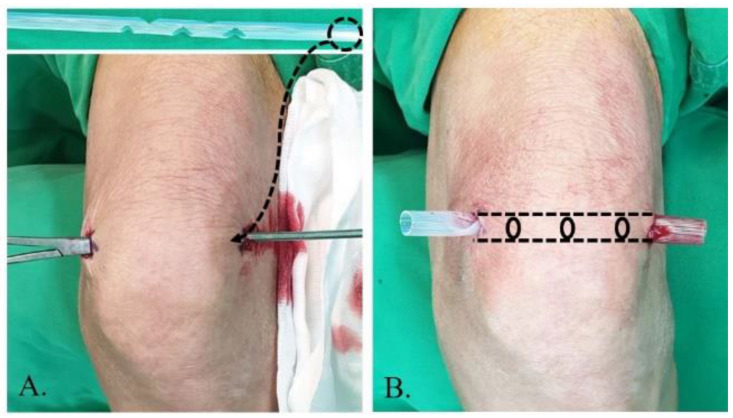
Tube insertion strategy from the superolateral to superomedial portal performed under local anesthesia. (**A**) The tube is inserted using Kelly forceps and switching stick. (**B**) The tube is positioned by tagging.

**Table 1 diagnostics-13-00371-t001:** Patient demographics.

	Group I (Arthroscopic Treatment)	Group II (Continuous Lavage)	*p* Value
Number of knees	76	34	
Re-aggravation	5 (IN: 4, OUT: 1)	1 (OUT: 1)	
Age, years	62.3 ± 11.6	66.1 ± 9.6	0.236
Sex, male/female	39%/61% (30/46)	40%/60% (8/12)	1.000
Side, right/left	47%/53% (36/40)	35%/65% (7/13)	0.161
BMI, kg/m^2^	23.7 ± 3.9	21.0 ± 3.7	0.551
Underlying disorder			
HTN	47% (36)	74% (25)	0.099
DM	21% (16)	26% (9)	0.220

BMI, body mass index; DM, diabetes; HTN, hypertension; IN, inpatient; OUT, outpatient.

**Table 2 diagnostics-13-00371-t002:** Laboratory results by study group.

	Group I	Group II	*p* Value
Initial serum			
WBC count (μL)	10825 ± 3560	9787 ± 2339	0.270
CRP (mg/dL)	15.0 ± 8.2	16.7 ± 9.1	0.473
Synovial fluid analysis			
WBC count (μL)	70934 ± 43322	71804 ± 41949	0.942
PMN (%)	85.0 ± 8.7	88.2 ± 5.2	0.061
Causative organisms			0.570
No growth	13% (10)	10% (4)	
Staphylococcus aureus	64% (49)	60% (20)	
Streptococci	8% (6)	9% (3)	
Staphylococcus epidermis	5% (4)	9% (3)	
Pseudomonas aeruginosa	4% (3)	6% (2)	
Enterococci	3% (2)	6% (2)	
Burkholderia cepacia	3% (2)	0% (0)	

CRP, C-reactive protein; PMN, polymorphonuclear cell; WBC, white blood cell.

**Table 3 diagnostics-13-00371-t003:** Patients’ variables by study group.

Patients’ Variables		Group I	Group II	*p* Value
Initial K-L grade		2.9 ± 0.6	2.6 ± 0.9	0.352
Ratio of initial K-L grade	Grade 1	3.9% (3)	11.8% (4)	0.125
	Grade 2	18.4% (14)	29.4% (10)	
	Grade 3	63.2% (48)	41.2% (14)	
	Grade 4	14.5% (11)	17.6% (6)	
Lavage number		1.1 ± 0.3 (2 times: 4, 3 times: 1)	1.1 ± 0.4 (2 times: 4)	0.952
Time to intervention (h)		23.6 ± 15.6	6.7 ± 9.3	<0.001 *
Hospitalization (days)		24.0 ± 16.2	24.7 ± 16.1	0.886
CRP normalization time (days)		21.7 ± 6.5	21.7 ± 7.0	0.981

CRP, C-reactive protein; K-L grade, Kellgren and Lawrence; * Statistical significance was set at *p* < 0.05.

**Table 4 diagnostics-13-00371-t004:** Patients’ clinical outcomes by study group.

	Group I	Group II	*p* Value
Range of motion			
3 months postoperative FC (N)			0.849
Normal	64.5% (49)	76% (26)	
<10°	32.9% (25)	24% (8)	
>10°	2.6% (2)	0% (0)	
3 months postoperative FF (N)			0.583
Normal	93.4% (71)	100% (34)	
Limited	6.6% (5)	0% (0)	
Clinical scores			
WOMAC pain			
Preoperative	12.3 ± 3.5	13.0 ± 3.8	0.442
3 months postoperative	2.4 ± 2.0	3.3 ± 3.0	0.186
WOMAC stiffness			
Preoperative	3.5 ± 2.4	3.9 ± 2.2	0.528
3 months postoperative	1.4 ± 1.5	1.3 ± 0.9	0.643
WOMAC function			
Preoperative	42.5 ± 10.9	39.4 ± 7.7	0.192
3 months postoperative	19.3 ± 8.3	16.1 ± 6.9	0.161

FC, flexion contracture; FF, further flexion; WOMAC, Western Ontario McMaster Universities Osteoarthritis index.

## Data Availability

Not applicable.

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
