# Peer review of "Efficacy of Lavage by Tube under Local Anesthesia versus Arthroscopic Treatment of Acute Septic Arthritis of Native Knee"

_diagnostics, 2023, doi:10.3390/diagnostics13030371_

Round 1
Reviewer 1 Report
English editing is required in the overall manuscript. In addition, considering that the procedure introduced by the author is limited to the knee joint, the contents of the entire manuscript should be modified.
Abstract
I think that arthroscopic treatment for septic arthritis is current treatment of choice, except for contraindication. It is not the past.
Arthroscopic treatment was effective for septic arthritis, but there were some problems because of general or 11 regional anesthesia. -> Although arthroscopic treatment is minimal invasive surgery and effective for septic arthritis of knee joint, it must be performed under general or regional anesthesia.
The purpose-> the purpose of this study
23.6±15.6 hour??
What are the “FF” and “FC”?
no difference in speed on CRP declination??
Introduction
Although, repeat arthroscopic treatment could be performed when recurrence of infection or aggravation occurs, it must be performed under general or regional anesthesia and this delays surgery.
Treatment efficacy??-> I recommend “the comparison of clinical result”.
Methods
Wasn't daily irrigation performed after 3L normal saline irrigation with the first tube insertion?
The daily management after tube insertion needs to be described in detail.
Results
” There were more re-aggravation cases in group I than group II.”
“There was no recurrence in group II.”
Hospitalization -> length of hospital stay
What are the “FF” and “FC”?
Discussion
The author's technique is to perform drainage with the joint cavity open, which I think is likely to be exposed to other infective orgarnism. A description of the pros and cons of the procedure described by the author and a comparison with other treatment methods will enrich the content of the discussion, especially, closed drainage.
Author Response
Thank you for your constructive comments on our manuscript Diagnostics (ISSN 2075-4418) Manuscript ID: diagnostics-2071104; Title: "Efficacy of Lavage by Tube on Local Anesthesia Compared with Arthroscopic Treatment of Acute Septic Arthritis of Native Knee"
We have tried our best to correct our manuscript as suggested by the reviewers. Specific details of the corrections are attached. Please find an attached file entitled ‘Response to reviewers’ letter. We have highlighted the revision file (track change) as you recommended and marked it as a yellow color.
< Point-by-point response to Evaluation >
Reviewer 1
English editing is required in the overall manuscript. In addition, considering that the procedure introduced by the author is limited to the knee joint, the contents of the entire manuscript should be modified.
Response: Thank you for your kind comments. We have made this manuscript read and corrected by a native English speaker. We performed 2 times English edition.
Abstract
I think that arthroscopic treatment for septic arthritis is the current treatment of choice, except for contraindication. It is not the past.
Response: Thank you for your kind comments.
Arthroscopic treatment was effective for septic arthritis, but there were some problems because of general or 11 regional anesthesia. -> Although arthroscopic treatment is minimal invasive surgery and effective for septic arthritis of knee joint, it must be performed under general or regional anesthesia.
Response: Thank you for your valuable comments. As your comments, we edited it. (Line 9-10)
The purpose-> the purpose of this study
Response: As your comments below, we edited it. (Line 10-11)
23.6±15.6 hour??
Response: Yes, sometimes we could proceed operation with no delaying. It was the average time to intervention. I described unit. (Line 19)
What are the “FF” and “FC”?
Response: Thank you for your valuable comments. We removed it for concise abstract.
no difference in speed on CRP declination??
Response: Thank you for your valuable comments. we added and edited descriptions more comprehensible at line 20
Introduction
Although, repeat arthroscopic treatment could be performed when recurrence of infection or aggravation occurs, it must be performed under general or regional anesthesia and this delays surgery.
Treatment efficacy??-> I recommend “the comparison of clinical result”.
Response: Thank you for your valuable comments. As your comments, we edited it. (Line 63-64)
Methods
Wasn't daily irrigation performed after 3L normal saline irrigation with the first tube insertion?
The daily management after tube insertion needs to be described in detail.
Response: Thank you for your comments. We did not irrigation daily. We added and edited some descriptions to explain (Line 119-120)
Results
” There were more re-aggravation cases in group I than group II.”
Response: Thank you for your comments. We added and edited some descriptions (Lines 155-156)
“There was no recurrence in group II.”
Response: Thank you for your comments. We removed it.
Hospitalization -> length of hospital stay
Response: We added and edited some descriptions (Line 93)
What are the “FF” and “FC”?
Response: Thank you for your valuable comments. As your comments, we explained abbreviation at line 95.
Discussion
The author's technique is to perform drainage with the joint cavity open, which I think is likely to be exposed to other infective orgarnism. A description of the pros and cons of the procedure described by the author and a comparison with other treatment methods will enrich the content of the discussion, especially, closed drainage.
Response: Thank you for your kind review. As your comments, we added other results about septic knee treatments and described them at line 261-280

Reviewer 2 Report
The authors investigated the “Efficacy of Lavage by Tube on Local Anesthesia Compared with Arthroscopic Treatment of Acute Septic Arthritis of Native Knee”.
The topic of this study is quite interesting and the decision of surgical treatment in acute septic arthritis is challenging due to the patient’s comorbidities and inaccurate diagnosis. In these circumstances, lavage by tube on local anesthesia might be a valuable tool in treating acute septic arthritis in the knee joint.
However, this article has several problems, so it should be corrected. Therefore, if appropriate revision is made, it will be possible to be published in DIAGNOSTICS.
Abstract
Line 18 Please provide how to evaluate the clinical outcomes.
Line 21 Address the unit of time to the intervention. (ex, day, hour)
Line 21 Please spell out abbreviations when they are first introduced. (FC and FF)
Line 22 Which differences was not significant? Differences between FC and FF?
Line 25 Please spell out abbreviations when they are first introduced. (POD)
Introduction
Line 71 There are three hypotheses in this study. Rephrase the sentence to plural.
Line 73 Is the third hypothesis the differences in functional outcomes between the two treatments? Please revise the third hypothesis.
Method
Line 76-77 Were all the patients between Jan 2019 and April 2022 treated with lavage by tube only? Are there any patients treated with arthroscopic surgery?
Line 79 This sentence is hard to understand. Please rephrase the sentence as follow; ‘The inclusion criteria were patients with septic arthritis of the knee joint as confirmed by laboratory markers in blood and joint fluid obtained by the arthrocentesis.’
Line 111 Please spell out abbreviations when they are first introduced. (FC and FF)
Line 117 Did all patients undergo arthroscopic lavage only? Or was synovectomy performed at the same time?
Line 120 The abbreviation POD was introduced above. Please remove the word ‘postoperative day’.
Line 161 K-L grade is an ordinal categorical variable. The categories in the categorical variable do have a natural order, but distances between categories cannot be quantified. Therefore, it should be compared using the chi-square test.
Line 192 Please spell out abbreviations when they are first introduced. (NRS)
Discussion
Line 258 The follow-up periods of the study were short-term at three months. This point should be discussed in the limitation.
Table 1
Line 173 Please change ‘site’ to ‘side’.
Line 173 The asterisk in the annotation should be addressed in the table. Please insert the asterisk (*) in the table.
Table 2
Line 183 The asterisk in the annotation should be addressed in the table. Please insert the asterisk (*) in the table.
Line 183 Please check the unit of WBC. (mm3)
Table 3
Line 197 What does the N in parentheses refer to?
Line 197 As mentioned above, K-L grade should be compare using the chi-square test.
Line 197 The annotation should be addressed in the table. Please remove WBC and PMN.
Author Response
Thank you for your constructive comments on our manuscript Diagnostics (ISSN 2075-4418) Manuscript ID: diagnostics-2071104; Title: "Efficacy of Lavage by Tube on Local Anesthesia Compared with Arthroscopic Treatment of Acute Septic Arthritis of Native Knee"
We have tried our best to correct our manuscript as suggested by the reviewers. Specific details of the corrections are attached. Please find an attached file entitled ‘Response to reviewers’ letter. We have highlighted the revision file (track change) as you recommended and marked it as a yellow color.
< Point-by-point response to Evaluation >
Reviewer 2
The authors investigated the “Efficacy of Lavage by Tube on Local Anesthesia Compared with Arthroscopic Treatment of Acute Septic Arthritis of Native Knee”.
The topic of this study is quite interesting and the decision of surgical treatment in acute septic arthritis is challenging due to the patient’s comorbidities and inaccurate diagnosis. In these circumstances, lavage by tube on local anesthesia might be a valuable tool in treating acute septic arthritis in the knee joint.
However, this article has several problems, so it should be corrected. Therefore, if appropriate revision is made, it will be possible to be published in DIAGNOSTICS.
Response: Thank you for your valuable comments.
Abstract
Line 18 Please provide how to evaluate the clinical outcomes.
Response: As your comments, we edited it. (Line 17)
Line 21 Address the unit of time to the intervention. (ex, day, hour).
Response: As your comments, we edited it. (Line 19)
Line 21 Please spell out abbreviations when they are first introduced. (FC and FF)
Response: We removed it for concise abstract.
Line 22 Which differences was not significant? Differences between FC and FF?
Response: As your comments, we removed that phrase.
Line 25 Please spell out abbreviations when they are first introduced. (POD)
Response: As your comments, we changed it at whole pages.
Introduction
Line 71 There are three hypotheses in this study. Rephrase the sentence to plural.
Line 73 Is the third hypothesis the differences in functional outcomes between the two treatments? Please revise the third hypothesis.
Response: Thank you for your comments. As your comments, we removed it for more comprehensible. It means that we tried to compare functional outcomes as well.
Method
Line 76-77 Were all the patients between Jan 2019 and April 2022 treated with lavage by tube only? Are there any patients treated with arthroscopic surgery?
Response: Thank you for your comments. Yes, we treat them only lavage by tube.
Line 79 This sentence is hard to understand. Please rephrase the sentence as follow; ‘The inclusion criteria were patients with septic arthritis of the knee joint as confirmed by laboratory markers in blood and joint fluid obtained by the arthrocentesis.’
Response: Thank you for your comments. As your comments, we changed it at lines 71-72.
Line 111 Please spell out abbreviations when they are first introduced. (FC and FF)
Response: Thank you for your comments. As your comments, we edited it at whole pages.
Line 117 Did all patients undergo arthroscopic lavage only? Or was synovectomy performed at the same time?
Response: Thank you for your comments. We added describes about synovectomy. (Line 109-110)
Line 120 The abbreviation POD was introduced above. Please remove the word ‘postoperative day’.
Response: Thank you for your comments. As your comments, we removed it
Line 161 K-L grade is an ordinal categorical variable. The categories in the categorical variable do have a natural order, but distances between categories cannot be quantified. Therefore, it should be compared using the chi-square test.
Response: Thank you for your comments. We tried to show that there was no difference between the two groups at K-L grade. As your comments, we edited it at Table 3 (Lines 207)
Line 192 Please spell out abbreviations when they are first introduced. (NRS)
Response: Thank you for your comments. As your comments, we described it at line 200.
Discussion
Line 258 The follow-up periods of the study were short-term at three months. This point should be discussed in the limitation.
Response: Thank you for your comments. As your comments, we described it at line 285-288
Table 1
Line 173 Please change ‘site’ to ‘side’.
Response: We changed it as your comments. (Line 165)
Line 173 The asterisk in the annotation should be addressed in the table. Please insert the asterisk (*) in the table.
Response: Thank you for your review. However, there were no applicable p-value.
Table 2
Line 183 The asterisk in the annotation should be addressed in the table. Please insert the asterisk (*) in the table.
Response: There were no applicable p-value.
Line 183 Please check the unit of WBC. (mm3)
Response: We used mm3 equal as . As your comments, we changed it. (Line 190)
Table 3
Line 197 What does the N in parentheses refer to?
Response: As your comments, we removed that misunderstandable phrase.
Line 197 As mentioned above, K-L grade should be compare using the chi-square test.
Response: As your comments, we edited it at Table 3 (Lines 207)
Line 197 The annotation should be addressed in the table. Please remove WBC and PMN.
Response: As your comments, we removed it and edited table.
